# Common evolutionary origin of acoustic communication in choanate vertebrates

Gabriel Jorgewich-Cohen [1] ✉, Simon William Townsend [2,3,4], Linilson Rodrigues Padovese [5], Nicole Klein [1], Peter Praschag[6], Camila R. Ferrara[7], Stephan Ettmar [8], Sabrina Menezes[9], Arthur Pinatti Varani[10], Jaren Serano[11] & Marcelo R. Sánchez-Villagra [1] ✉

Acoustic communication, broadly distributed along the vertebrate phylogeny, plays a fundamental role in parental care, mate attraction and various other behaviours. Despite its importance, comparatively less is known about the evolutionary roots of acoustic communication. Phylogenetic comparative analyses can provide insights into the deep time evolutionary origin of acoustic communication, but they are often plagued by missing data from key species. Here we present evidence for 53 species of four major clades (turtles, tuatara, caecilian and lungfish) in the form of vocal recordings and contextual behavioural information accompanying sound production. This and a broad literature-based dataset evidence acoustic abilities in several groups previously considered non-vocal. Critically, phylogenetic analyses encompassing 1800 species of choanate vertebrates reconstructs acoustic communication as a homologous trait, and suggests that it is at least as old as the last common ancestor of all choanate vertebrates, that lived approx. 407 million years before present.

Despite the unquestionable importance of acoustic communication among vertebrates, our knowledge regarding its origin remains sparse. The current consensus based on available evidence favours a convergent origin of acoustic communication among vertebrates: studies on acoustic sensory abilities show that the morphology in the hearing apparatus and its sensitivity vary considerably among vertebrates[1–3]. This, in addition to observed differences in vocal tract morphology, suggests that acoustic communication likely evolved multiple times, emerging independently among diverse clades[3]. Phylogenetic analyses used to reconstruct the ancestral state of acoustic communication along the tree nodes, whilst suggestive of multiple origins[4], are arguably complicated by missing data from key taxa.

An alternative hypothesis is that acoustic communication has a common and ancient evolutionary origin. In support of this, vertebrate hearing epithelia and cerebral promotor circuits that control vocal behaviours are considered to be homologous and operate in the same hindbrain compartment, respectively[5–9]. Furthermore, in spite of the variety of sound production mechanisms, all Choanata (Dipnoi (lungfishes) + Tetrapoda) lineages have lungs as the physical source of their calling behaviours.

Among vertebrates, clades that can be easily recognised to produce complex sounds (i.e. frogs, crocodilians, birds and mammals) have been studied extensively (e.g. ref. 10–12). However, some vertebrate clades, in contrast, have been assumed to be non-vocal based on limited or sparse data. As a consequence, the absence of concrete

[1]Paleontological Institute and Museum, University of Zurich, Zurich, Switzerland. [2]Center for the Interdisciplinary Study of Language Evolution, University of Zurich, Zurich, Switzerland. [3]Department of Comparative Language Science, University of Zurich, Zurich, Switzerland. [4]Department of Psychology, University of Warwick, Coventry, UK. [5]Department of mechanical engineering, University of Sao Paulo, Sao Paulo, Brazil. [6]Turtle Island – turtle conservation and research center, Graz, Styria, Austria. [7]Wildlife Conservation Society – WCS Brasil, Manaus, Amazonas, Brazil. [8]ZooCon zoological consulting, Neudörfl, Burgenland, Austria. [9]Federal University of Tocantins, Palmas, Tocantins, Brazil. [10]Sao Paulo Metodista University, Sao Bernardo do Campo, Sao Paulo, Brazil. [11]Department of Wildlife Ecology and Conservation, University of Florida, Gainesville, FL, USA. ✉e-mail: gabriel.jorgewichcohen@pim.uzh.ch; m.sanchez@pim.uzh.ch

evidence for vocal production is sometimes treated as evidence of non-vocal tendencies (e.g. ref. 4). Central to a robust reconstruction of acoustic communication is a systematic documentation of these key, neglected groups.

Here, we investigate the evolutionary origins of acoustic communication in choanate vertebrates combining critical data with phylogenetic trait reconstruction methods using a comprehensive dataset. We assess the acoustic communication abilities in species of diverse vertebrate groups, including Lepidosauria (tuataras, lizards and snakes), non-anuran Amphibia (salamanders and caecilians), Chelonians (turtles) and lungfishes (Dipnoi) that are key to mapping vocal communication in the vertebrate tree of life. Using this dataset combined with data of well-known acoustic clades (e.g. mammals, birds and frogs), we test if the evolutionary origin of acoustic communication is shared among choanate vertebrates. We suggest a single origin of acoustic communication in the last common ancestor of all Choanata over 400 million years before present (mybp).

## Results

### Origins of acoustic communication

We found widespread evidence for acoustic behaviour among all choanate vertebrates. Our recordings include 53 species that belong to groups often thought to be non-vocal and commonly neglected in vocal communication research (Supplementary data 1). Of these, 50 species are turtles—representing over 54% of all genera and more than 14% of all extant species[13]. We also recorded tuataras (*Sphenodon punctatus*), one species of caecilian (*Typhlonectes compressicauda*), and the South American lungfish (*Lepidosiren paradoxa*). All recorded species were found to possess a varied acoustic repertoire comprising

a number of different sounds (see Figs. 1, 2 and supplementary Data 2 to listen to sounds, and supplementary Data 3 for sound descriptions).

A critical review of the extensive literature focusing on groups often considered to lack acoustic communication resulted in a total of 106 species, including 54 turtles, 14 lepidosaurs, 29 salamanders, four caecilians, four frogs and one lungfish having been reported to engage in vocal communication (Supplementary Data 1).

The African lungfish (*Protopterus annectens*) has been reported to produce sounds[14] and to being able to perceive sounds both in the water and air[15]. Among the ten families of caecilians, we found evidence of acoustic communication in representatives of four of them (Dermophiidae, Grandisoniidae, Ichthyophiidae and Siphonopidae)[16-19]. In salamanders, eight out of 10 families have representatives known to produce vocalisations, with evidence being absent only in Hynobiidae and Rhyacotritonidae[18-22]. We also found evidence for acoustic communication in all species of frogs of the genera *Ascaphus*[23,24] and *Leiopelma*[24-26]. Among Lepidosauria, examples of acoustic communication are found in most groups of Gekkota[27,28], and in tuataras (ref. 29; present study). Among turtles, we found evidence of acoustic communication in representatives from all families, with some species producing over 15 different types of calls used in various situations, including parental care[30,31]. These findings confirm that the ability to produce vocalisations is distributed across such groups.

### Ancestral-state reconstructions

The ancestral-state reconstruction for Choanata recovered the presence of acoustic communication as an unambiguous homologous trait, being present in the common ancestor of all choanate vertebrates (407 mybp), and in the majority of the tree nodes (e.g. tetrapods, amniotes, reptiles; Fig. 1). An ancestral-state reconstruction

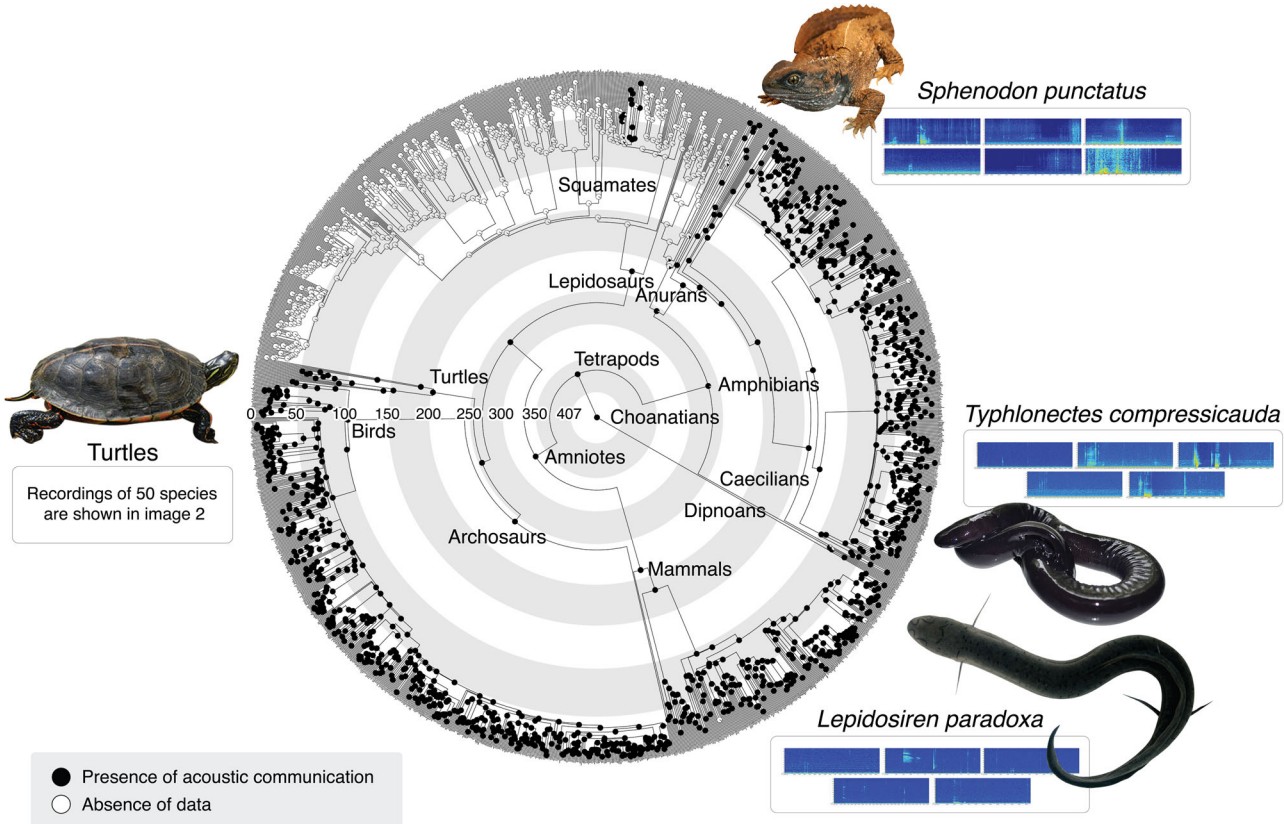

**Fig. 1 | Choanate vertebrates' acoustic communication ancestral state reconstruction analysis.** Tree includes 1800 choanatian species assigned either with the character presence or absence of acoustic communication. Pie charts at ancestral nodes indicate likelihood of each character state. Colours in the spectrograms represent sound intensity, with warm colours representing high intensity and cold colours (i.e. blue) representing low intensity or absence of sounds. Character states for each species can be accessed in Supplementary Data 4.

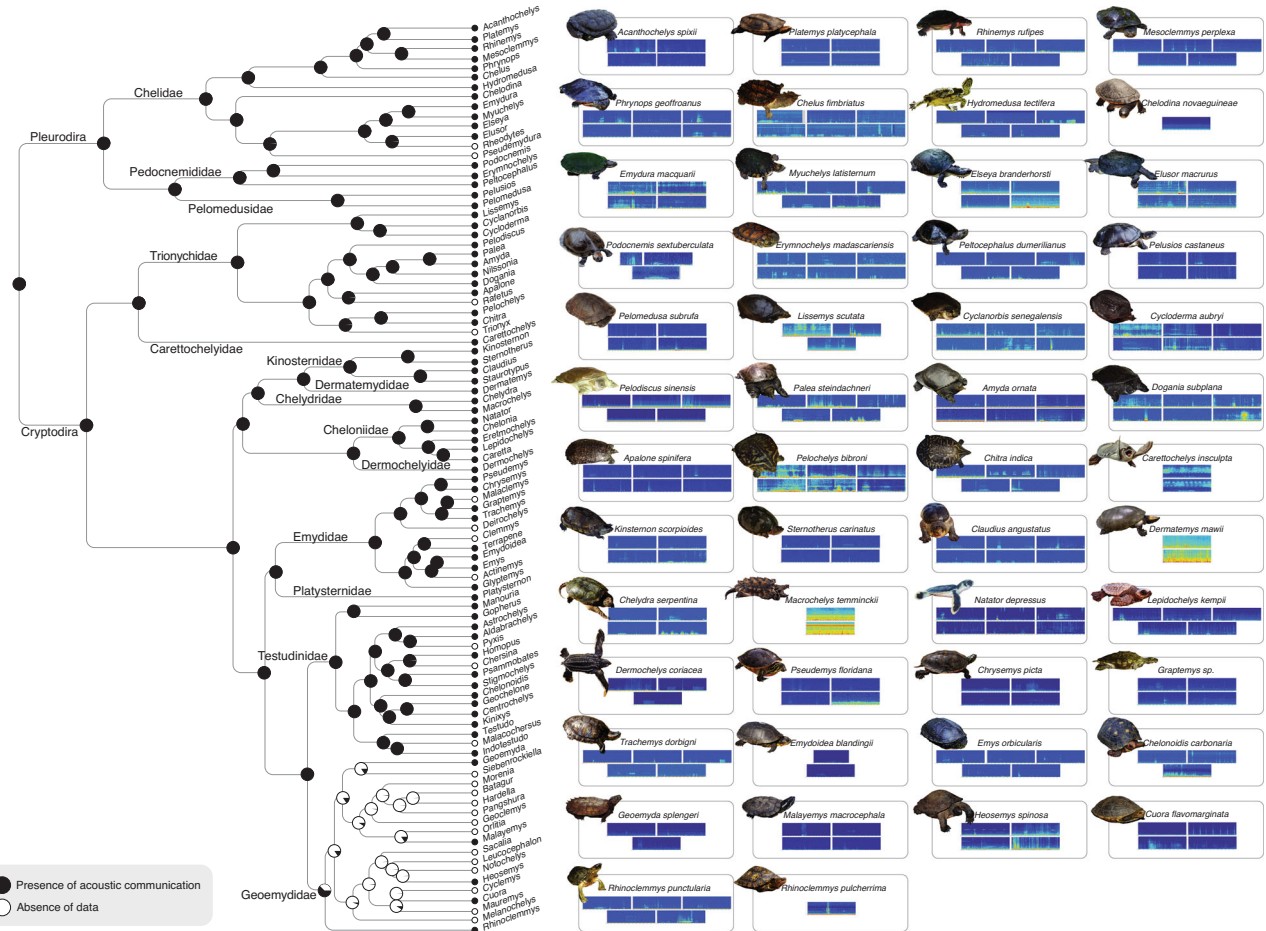

**Fig. 2 | Turtle acoustic communication ancestral-state reconstruction analysis.** Tree includes every turtle genus assigned either with the character presence or absence of acoustic communication. Pie charts at ancestral nodes indicate the likelihood of each character state. Colours in the spectrograms represent sound intensity, with warm colours representing high intensity and cold colours (i.e. blue) representing low intensity or absence of sounds. Character states for each species can be accessed in Supplementary data 5.

using a tree containing only turtle genera resulted in the presence of acoustic communication in every ancestor node, except for the Geoemydidae family (Fig. 2). This is likely an artefact of the effect of missing data given the limited representation of species of this clade in our dataset, many of which are endangered and hard to access. Nevertheless, the presence of this trait is unambiguously ancestral among turtles.

## Discussion

Data across the turtle tree of life together with other critical taxa in combination with available evidence across all major tetrapod clades totalling in excess of 1800 species confirms a common origin of sound production and acoustic communication among choanate vertebrates, dating from the Palaeozoic (at least 407mybp). These findings support the hypothesis that innovations in the sound production apparatus among choanates were acquired after the first, common appearance of acoustic communication within this group.

The interpretation of acoustic behaviour as a non-homologous trait proposed in previous research[3,4] was driven largely by a lack of information on key groups of animals. That is, analyses of ancestral-state reconstruction are complicated by missing data which can subsequently be treated as evidence of absence. Nevertheless, the recent growth of evidence for acoustic communication among certain tetrapod groups, commonly considered to be non-vocal, such as aquatic turtles (e.g. ref. 31,32, and the data provided by us in this paper), are key in revealing the common ancestry of such behaviour. In fact,

including evidence from only 14 species (12 turtles, tuatara and lungfish) to the analysis proposed by Chen & Wiens[4] was enough to recover opposite results, that were reinforced by the inclusion of data from our critical study of the literature. The sensitivity of ancestral state reconstruction analyses to the character state of key lineages makes a deeper investigation of poorly studied groups imperative.

Knowledge of the natural history of organisms is fundamental in surveys of the macroevolution of certain features. The intensive documentation of vocal communication in turtles in our study is an example. Recordings, observations and subsequent analyses in a phylogenetic framework suggest the homology of vocal communication across turtles and in the last common ancestor of the clade. This result strengthens our broader conclusions on the origins of acoustic communication among choanate vertebrates. Specifically, by evidencing that acoustic communication is widespread and homologous among all turtle genera, we ensure that analysis with a much less comprehensive sample of this group is trustworthy and not subject to extensive interpretation changes by switching the state character of only a few species.

With the inclusion of several taxa of comparatively understudied groups to the analyses, we show that the use of sounds in communication is not only a homologous but also conserved behaviour, widely distributed among choanate vertebrates. The wide variety of mechanisms of sound production—with some of the most distinguished examples being the bird syrinx[33,34], the trombone-like crest of *Parasaurolophus* dinosaurs[35,36], and the sound apparatus of bats and

dolphins, capable of producing ultrasounds[37]—also deserves consideration, in order to reveal the anatomical and physical transformations that must have been involved around a common neurobiological framework of sound production.

The larynx has an important role in tetrapod acoustic production, being the main site of vocal production in most lineages[38]. Although some of the acoustic structures used by choanate vertebrates do not share their embryological origins with the larynx—for example, the bird syrinx develops from different tissues[33,34,39]—they all share the use of air circulation in the production of sounds, powered by the lungs (with the exception of Plethodontid salamanders that lost the latter). Additionally, all choanate vertebrates are able to produce laryngeal sounds, including birds, while hissing[40].

Furthermore, all vertebrates share the location of motoneurons associated with vocalisations in the caudal hindbrain[5,41]. Vocal-acoustic and pectoral-gestural signalling also share evolutionary and developmental neural origins[5,9,42], which implies a common vocal-sonic central pattern generator in the vertebrate brain[5,9,41,43].

Many salamanders, caecilians, lizards and snakes produce complex, modulated sounds[44,45], that were not considered acoustic communication in the previous analysis due to its usage being usually applied to inter-specific communication[4]—e.g. defensive behaviour[44,46]—or allegedly produced by accident[47]. However, it cannot be excluded that these sounds could have had a common evolutionary origin to those used for intraspecific communication. The same rationale applies to simpler sounds. Hissing and sniffing sounds produced by most vertebrates, especially amniotes[48]—and nearly all reptiles[28]—might also contain more information than what we account for. e.g. non-vocal sounds encode individual signals in birds[40] and some colubrid snakes mimic the hiss of vipers[49].

Considering the fact that communication is usually multimodal, and the loss of acoustic abilities can quickly happen due to redundancy with other channels such as visual and chemical ones in many taxa[50], analyses based on ancestral state reconstruction can be biased due to character interpretation and loss of track caused by recent changes in character state in the tip species. The widespread usage of hissing and sniffing sounds among vertebrates might be further evidence that acoustic communication is a shared character in this group and started before the diversification of sounds and their various usages by different clades. Because we chose more conservative methods that include only sounds that play a role in communication with conspecifics, and excludes inter-specific communication (e.g. defence hisses by snakes and other species), future studies might broaden this scope by including such calls, since hisses, in particular, appear to be common across tetrapods, and are typically defence vocalisations.

If all sounds have the same evolutionary origin, vocalisations produced by clades commonly conceived to have had a secondary development of acoustic communication in previous works, such as non-anuran amphibians and Gekkota, mainly Gekkonidae and Pygopodidae[4,28], would be homologous to the ones produced by other vertebrates.

The recovery of a single origin of acoustic communication among vertebrates in our analysis reinforces the need to investigate hissing and sniffing sounds, here not classified as acoustic communication. Investigations on the potential primary homology of these sounds, taking a neurobiological, physiological, and anatomical approach, would be important to shed light on this discussion. Comparative studies on the diversification of calls and vocal apparatus, including different vertebrate groups, are also needed to clarify the question of homology.

Vocalisations are also a widespread behaviour among actinopterygian fishes. In this case, however, it may have evolved recurrently and independently over 30 times during their evolutionary history[5,9,51–53]. Although many are the mechanism of sound production among actinopterygians, swim bladder vibrations seem to be the most widespread and ancient of them[53]. Considering that the homology of lungs among vertebrates is still debated, with strong evidence for a common origin between the lung and the swim bladder[54–56], a shared origin of acoustic behaviour between choanates and actinopterygians cannot be ruled-out. In this case, expanding our analysis to include actinopterygians may reveal the origins of acoustic communication to be even older than 407mybp[57].

Whilst inferring a common origin of acoustic communication among actinopterygians and choanates may be complicated at present due to lack of data, additional alternative evidence points in the same direction—perhaps a deep homology[58] of shared brain mechanisms[5,9,41–43]. Furthermore, the gill arches used by fish as a breathing system evolved into the tetrapod hyoid and laryngeal apparatus, used in many mechanisms that include breathing, feeding and sound production[59]. Such connections might be modulated by the same or similar brain channels, that might suggest evolutionary continuity[9].

A challenge to test the hypothesis of a common origin of acoustic communication among some actinopterygian lineages and choanates will be understanding how the morphological transformations involved in the transition to choanates affected the mechanisms of sound production. Palaeontological data and phylogenetic reconstructions are the most common approaches applied to shed light on the evolutionary transformation of traits. However, to date, palaeontological data that are key to a robust resolution of the origins of acoustic communication among actinopterygians and sarcopterygians are missing[48,60]. This raises a number of additional questions, but of particular importance: is the acoustic communication of choanates an innovation homologous to the acoustic communication based on the swim bladder observed in fishes? If so, did acoustic communication first appear among actinopterygians or in the event that precedes them—such as in the transition from 'protochordates' (-550 mybp, 5)? Integrative comparative studies of embryological development, physiology of vocal apparatus and of vocal neural architecture across brain regions combined with gene expression among taxa will be helpful to trace the evolution of acoustic behaviour among vertebrates. The common ancestral origin of acoustic communication provides further justification for the use of choanate animals as models in the study of the origins of human language and speech.

## Methods
No general ethics approval was required for this study as it was conducted with animals that were already captive. Nevertheless, approval was granted by specific institutions that required analysis from a committee (includes Chester Zoo, CEQUA and Turtle Island).

### Sound recordings and analysis
For underwater sound recordings, we used the OceanBase (developed by the Laboratory of Acoustic and Environment—University of Sao Paulo, in partnership with Bunin tech®), an acoustic recorder specifically designed for underwater noise monitoring. It has a sensitivity of $-157 \pm 2$ dB rel 1 V/uPa $\pm 2$ dB and a frequency band of 5 Hz–90 kHz. In-air recordings were conducted using a Tascam® recorder DR-100MKIII with a sensitivity of $-115.5 \pm 0.5$ dB rel 13 mV/uPa $\pm 4$ dB and a frequency band of 5 Hz–96 kHz.

Recordings were made in captivity using plastic pools to ensure that all sounds were produced by the animals being recorded. Each species was recorded for at least 24 h, capturing both day and night activity. We aimed for recording males and females in different life stages whenever specimens were available. We also recorded ambient sound without the presence of any animals in order to account for possible noise/interference.

Sounds recorded were analyzed using Raven Pro 1.6[61] and Praat[62]. Sounds were measured using six acoustic parameters: Number of bouts, fundamental frequency (Hz), minimum frequency (Hz), peak frequency (Hz), duration (s), and sound type (simple or complex).

These parameters were only used as a first description of the repertoires, and were not subjected to any analysis in the present study.

## Acoustic communication data

We recorded 53 species, that include 50 turtle species, one caecilian, one lungfish and tuataras, of which all communicated acoustically (Figs. 1, 2, Supplementary Data 1–3 and Supplementary Note 1). Apart from the vocalisations we recorded, most of the acoustic communication data used in this work originates from the dataset published by Chen & Wiens[4], that includes 1799 tetrapod species (supplementary Data 4). In addition, we searched for information on acoustic communication among groups that are often considered to be silent (i.e. Testudines, Lepidosauria, Gymnophiona, Caudata and some anuran species). We gathered information from peer-reviewed articles, books and personal communication with researches that work directly with the referred groups (Supplementary Data 1). Our search was conducted using Google Scholar and Web of Science between March and November 2021 using the keywords "acoustic communication", "call", "vocal communication", "vocalisation", "song", and "sound", in association with the species' name and superior taxonomic ranks. The search was conducted following PRISMA[63] guidelines (Supplementary file 1).

Sound communication entails not only that the animal is producing a sound, but also that it has communicative significance. To avoid mistakes in determining its significance, and to ensure we are comparing homologous types of acoustic communication, we favoured the hypothesis that the presence of a complex repertoire (presence of a number of different sounds and/or harmonic calls) entails communicative meaning and considered only sounds produced by the respiratory tract (excluding scale scratching and tail rattling, for example). We also decided to exclude lungless salamanders of the Plethodontidae family, as they might have a different, non-homologous method of sound production[20].

Additionally, in order to ensure character homology, our analysis includes only Sarcopterygian lineages (namely Choanata: Tetrapoda + Dipnoi; ref. 64), as we hypothesised the presence of lungs as a major driver for acoustic behaviour in this clade. Sound production systems in other vertebrate clades are diverse and we have not yet enough evidence to infer its homology.

Sounds produced during defensive behaviour such as hissing and sniffing in lizards or bellowing in snakes were not considered to be intraspecific acoustic communication and, therefore, were not included. Although these behaviours might have a common origin to the sounds here considered acoustic communication, we lack evidence to support this claim and opted for a more conservative approach. Only sounds that were considered by literature reports to be used by the studied species in intraspecific communication were considered.

Compiled information regarding amphibians and reptiles that, in discordance with common beliefs, are capable of producing sounds, were compiled (Supplementary Data 1).

## Phylogenies

Two different phylogenies were used in this work. In order to analyze the origins of acoustic communication among choanate vertebrates, we used the tree from ref. 57, modified by ref. 4. Besides including representatives of all clades of tetrapods in the family level in a proportional sampling, it matches the available information on acoustic communication for 1799 species[4]. We made a minor modification to the tree by including Dipnoi as the extant outgroup to Tetrapoda (Choanata). We used the lungfish (*Lepidosiren paradoxa*) as the sister taxon to all tetrapods and inserted a branch length of 407mybp, based on *Eoactinistia foreyi*, the oldest coelacanthimorph, from the Devonian[57,65].

Although the position of turtles among reptiles is still debated[66–68], we decided to position it as the sister taxa to archosaurians, as most commonly recovered by recent molecular work[69,70].

In the second analysis, we used the turtle phylogeny proposed by Pereira et al.[71]. This is not the most recent phylogeny available[72], but it is the one with the largest overlap with our dataset. In any case, the relationship among genera is the same in both trees. We used the function drop.tip from the Ape package[73] in R platform[74] to exclude terminals. A tree containing each living turtle genus was created and used to analyze the distribution of sound production among turtles.

## Ancestral-state reconstruction

We based our analysis for choanate vertebrates on the dataset compiled by ref. 4. We reassigned character states based on the information gathered in our literature search and our own recordings: 0 for the absence of acoustic communication (which is, in many cases, no more than the absence of information) and 1 for presence. The same analysis was used for the turtle genera tree. Character states assigned to each species can be found in Supplementary Tables 5, 6, respectively.

Considering the great amount of missing data regarding turtle vocal behaviours, we inferred the presence of acoustic communication to a genus whenever at least one of its representatives is known to do so. The evolution of acoustic communication was inferred for each ancestral node across-tree using maximum-likelihood reconstruction, and the equal-rates model (ER).

## Reporting summary

Further information on research design is available in the Nature Research Reporting Summary linked to this article.

## Data availability

The authors declare that the data supporting the findings of this study are available within the paper and/or its supplementary information files: Supplementary Data 1 contains the list of species and corresponding sources obtained in the literature search. Sound repertoires of the species recorded in the present work can be found in Figs. 1, 2 and in detail in Supplementary Data 2 or in an online interactive presentation (shorturl.at/cwMU2). Supplementary Data 3 contains the description of the repertoires and the conditions in which each species was recorded. Supplementary Note 1 includes the PRISMA workflow with the methods used in the literature search, together with the resulting list of references. The literature search was conducted through the platforms Web of Science (https://clarivate.com/webofsciencegroup/solutions/web-of-science/) and Google scholar (https://scholar.google.com/). Supplementary Data 4, 5 contain the character coding information used in the choanate and the turtle analyses, respectively. Code and input files can be found in Supplementary Code 1.

## Code availability

This study used available R packages rather than custom coding. Nevertheless, we provide code and input files in the supplementary material. All code used for this work is provided in Supplementary Code 1.

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

## Acknowledgements
Swiss Government Excellence Scholarship (ESKAS) supported G.J.-C. (Grant number: 2020.0190). This work was supported by SNF Grant No. 31003A-169395 to M.R.S.-V. and by the Federal Commission for Scholarships for Foreign Students (FCS, Switzerland) to G.J.-C.

We thank Belize Foundation for Research and Environmental Education (BFREE) and Patrik Viana for *Dermatemys* and *Typhlonectes* photos, respectively; We thank Chester Zoo—Gerardo Garcia and the herpetology team, Turtle Island—Specially Maddy Wheatley and Lisa Marschnig, Centro de Estudos de Quelônios Amazônicos (CEQUA), Jansen Zuanon, Patrik Viana and Danilo Castanho for providing access to specimens. We also thank Dr. Torsten Scheyer and Lucas Sacheli Santos Parra for revising the manuscript and helping with the figures, respectively.

## Author contributions
All Authors gave substantial contributions to the present manuscript. G.J.-C., S.W.T. and M.R.S.-V. delineated the research; L.R.P. developed the recording equipment; G.J.-C., L.R.P., A.P.V., N.K., P.P., S.M., C.R.F., J.S. and S.E. collected and analyzed sound recordings; G.J.-C. performed the coding and statistical analyses; G.J.-C., S.E., M.R.S.-V. and S.W.T. contributed with text writing; and all authors contributed to the final revision.

## Competing interests
The authors declare no competing interests.
