## [Peer Review File · Nature Communications]

Common evolutionary origin of acoustic communication in choanate vertebratesReviewers' Comments:

Reviewer #1:

Remarks to the Author:

This manuscript is an interesting study that incorporates taxa that are generally neglected in the field of acoustic communication to reconstruct the evolution of acoustic behaviors among choanate vertebrates. The methods used are appropriate, the figures are clear and informative, and the study represents an important contribution to the field. I have a few comments that should be addressed in the revisions to this manuscript.

My major concern is that the character states used for the ancestral state reconstruction must be revised to be consistent with the authors' definition of acoustic communication (lines 311-319). Here, the authors indicate that they are considering acoustic signals for which sound energy is generated in the respiratory tract - if this is the case, they should code all species of the salamander family Plethodontidae as not having acoustic communication and revise lines 86-88 of the manuscript. In any case, most reports of sound production in salamanders remain fairly anecdotal (no data on call types or call parameters) and may more similar to the non-vocal acoustic behaviors described in lines 321-323 as being excluded from this study. It would help if the authors are more clear about their criteria for character state coding to accommodate the diversity of acoustic signal types included in this study. There are also several species that are notably absent that the authors might consider including in this study, particularly squamates for which evidence of acoustic repertoires have been recorded (e.g., *Liolaemus chiliensis*, Labra et al. 2013; *Anolis grahami*, Milton & Jenssen 1979).

Additionally, the manuscript does not make clear the purpose of the second analysis focusing on the turtle genera. How does this analysis address the main hypothesis of the study (the evolution of acoustic communication among choanates), and how does it add to the conclusions drawn using the larger phylogeny with greater taxon sampling?

Given that the vocalization data recorded in the present study are analyzed more in-depth (e.g., with call repertoires fairly well-defined in the supplement), the authors might consider mapping additional character traits onto this phylogeny that more fully characterize the richness of the vocal repertoire.

Regarding the methods: are recordings made in-air or under-water? The only recording equipment described is Oceanbase, an underwater recorder. There is no information for microphones used to record in air from terrestrial species (notably *Sphenodon punctatus*). Please also provide the dimensions of the plastic pools used for recording, describe any measures used to reduce generation of standing waves and/or noise if used, and indicate the distance between the source and the microphone.

The sound analysis described in the methods section (lines 292-296) does not appear to be included in the manuscript results or discussion. How many individuals per species were recorded, and how many calls per species (and per call type) were analysed? Several species included here appear not to have a repertoire of call types (e.g., *Rhinoclemmys pulcherrima* and *Chelodina novaeguineae* have only one call type characterized), and therefore (per the authors' definition of acoustic communication in lines 311-323) should be coded as 0 in the character map.

The years for the Wever citations refer to reprints, and should be corrected to reflect the original years of publication (1985 for *The Amphibian Ear*, 1978 for *The Reptile Ear*).

Reviewer #2:

Remarks to the Author:

Review of Jorgewich-Cohen et al. "Common evolutionary origin of acoustic communication in

vertebrates”

This paper represents an excellent idea, a valuable dataset, and a very worthwhile project overall, and I definitely think that with some revisions it should be published in Nature Communications.

However, although there are a number of rather minor changes in wording and focus, there are a few important oversights in the literature review, especially concerning snakes and crocodilians, that will require some additions to the database and thus re-analysis. I also think that the current Discussion is mostly tangential to the main findings of the study, and needs to be re-written.

I am somewhat reluctant to call these “minor revisions”, but will do so in the expectation that the authors will take the advice given below. I am confident that these additions will clearly strengthen the central conclusion: that tetrapod vocal communication represents a homologous trait, present in the common ancestor of all tetrapods (including sarcopterygian fish).

Major issues:

The discussion and dataset are conspicuously “turtle-centric” and leave out some of the other non-avian reptiles.

1. One group that is conspicuously absent from any mention in the ms. are the crocodilians, which are well-known to have diverse vocalizations and have been well-studied. The database does include Alligator mississippiensis as a vocal species (scored 1 in the supplemental dataset) but many other crocodilian species are also highly vocal and these need to be added for completeness. See:

Dinets, V. (2011). The Role of Habitat in Crocodilian Communication. (PhD). University of Miami, Miami, Florida.

Garrick, L. D., & Lang, J. W. (1977). Social signals and behavior of adult alligators and crocodiles. *American Zoologist*, 17, 225-239.

Reber, S. A., Nishimura, T., Janisch, J., Robertson, M., & Fitch, W. T. (2015). A Chinese alligator in heliox: formant frequencies in a crocodilian. *Journal of Experimental Biology*, 218, 2442-2447.

Vergne, A. L., Avril, A., Martin, S., & Mathevon, N. (2007). Parent-offspring communication in the Nile crocodile *Crocodylus niloticus*: do newborns' calls show an individual signature? *Naturwissenschaften*, 94(1), 49-54.

Vergne, A. L., Pritz, M. B., & Mathevon, N. (2009). Acoustic communication in crocodilians: from behaviour to brain. *Biological Reviews*, 84(3), 391-411.

2. As far as I can tell, all the snakes (Serpentes) are currently scored 0 in the current database, but there are various snake species that make vocalizations, including hisses (which are probably universal) and a few that make more interesting tonal calls. See:

Young, B. A. (1991). Morphological basis of "growling" in the king cobra, *Ophiophagus hannah*. *Journal of Experimental Zoology*, 260, 275-287.

Young, B. A., Sheft, S., & Yost, W. (1995). The morphology of sound production in *Pituophis melanoleucus* (Serpentes, Colubridae) with the first description of a vocal cord in snakes. *Journal of Experimental Zoology*, 273, 472-481.

Young's 1991 paper reports 21 snake species producing sounds. Interestingly, although the pine snake in the 1995 paper does use a larynx, the “vocal cord” equivalent Young reports looks like a novel innovation: relevant to my suggestions concerning the Discussion

3. The current Discussion seems quite tangential to the actual data reported here: it focuses almost entirely upon what was NOT studied (namely actinopterygian fish acoustic communication) and very little on the groups actually covered in the database. This needs to be rethought, and rewritten to focus on the data presented in the current study.

In particular, some discussion of the larynx as the (homologous) site of vocal production in all non-avian tetrapods, and at least a sentence or two about the syrinx as an avian “key innovation,” seem mandatory, and should replace some of the current speculation. The larynx is interesting as it is the main site of vocal production in tetrapods (with birds and odontocete cetaceans being major exceptions), but it also has some key innovations in mammals (the presence of a thyroid cartilage):

Fitch, W. T. (2016). Vertebrate bioacoustics: Prospects and open problems. In R. A. Suthers, W. T. Fitch, A. N. Popper, & R. R. Fay (Eds.), *Vertebrate Sound Production and Acoustic Communication* (pp. 297-328). New York: Springer.

Clarke, J. A., Chatterjee, S., Li, Z., Riede, T., Agnolin, F., Goller, F., . . . Novas, F. E. (2016). Fossil evidence of the avian vocal organ from the Mesozoic. *Nature*. doi:doi:10.1038/nature19852

Goller, F., & Larsen, O. N. (1997a). A new mechanism of sound generation in songbirds. *Proceedings of the National Academy of Sciences*, 94(26), 14787.

The current Discussion goes offers speculations about possible common origins between fish and other vertebrates. I think this would be fine as a final paragraph (in a “for future work” vein) but not dominating the whole discussion.

I would however note in this regard that the rhombomeric similarities cited by Bass in the cited papers across quite different vocal production systems, include the avian syrinx, which is well-known to be a novel organ and bird innovation. Thus, this similarity is weak evidence for any homology between tetrapods and fish.

4. The claim that turtles or salamanders are “traditionally” thought to lack vocal communication seems overstated.

Regarding turtles first, such references as Auffenberg’s review (not currently cited) make clear that turtles are vocal, though perhaps less so than other reptiles.
Auffenberg, W. (1977). Display behavior in tortoises. *American Zoologist*, 17, 241-250.

See also Gans & Maderson’s well-cited review, who clearly note vocalizations in turtles and snakes:
Gans, C., & Maderson, P. F. A. (1973). Sound producing mechanisms in recent reptiles: Review and comment. *American Zoologist*, 13, 1195-1203.

Therefore I’m not sure to what “tradition” the authors refer, but I’d argue that any well-informed bioacoustician is well aware that turtles make sounds (incidentally, so would any reader of D. H. Lawrence’s famous 1921 “Tortoise Poems” which describe how he thought tortoises were mute, until hearing them vocalize!)

The authors should either give some citations to a well-respected source (e.g. textbooks) to support the use of this term, or replace it with “commonly” or “often” thought...

Similar comments could be made regarding salamander voices – there is plenty of previous literature here e.g. reviews in:

Maslin, T. P. (1950). The production of sound in caudate amphibia. *University of Colorado Studies*, 1, 29-45.

Neill, W. T. (1952). Remarks on salamander voices. *Copeia*, 1952, 195-196.

Taxon-specific reports:

Coleman, A. (2016). Sound production in the small-mouthed salamander (*Ambystoma texanum*) (Masters).

Crovo, J. A., Zeyl, J. N., & Johnston, C. E. (2016). Hearing and Sound Production in the Aquatic

Salamander, *Amphiuma means* *Herpetologica*, 72(3), 167-173.

Davis, J. R., & Brattstrom, B. H. (1975). Sounds produced by the California newt, *Taricha torosa*. *Herpetologica*, 31, 409-412.

Weber, E., & Schumacher, R. (1975). Der Aufbau der Abwehrrufe des Kammmolches (*Triturus cristatus*) und des Fadenmolches (*Triturus helveticus*) (Amphibia, Caudata, Salamandridae). *Salamandra*, 11, 119-129.

Weber, E., & Schumacher, R. (1976). Verschiedenartige Abwehrrufe des Teichmolches (*Triturus v. vulgaris*) (Amphibia, Urodela). *Biologisches Zentralblatt*, 95, 693-701.

Wyman, R. L., & Thrall, J. H. (1972). Sound production by the spotted salamander *Ambystoma maculatum*. *Herpetologica*, 28, 210-212.

5. There is one published mention of vocal production in African lungfish, along with the South American lungfish, so they could be added to the list:

M'Donnell, R. (1860). Observations on the habits and anatomy of the *Lepidosiren annectans*. *Natural History Review*, 7, 93-112.

Minor corrections:

1. Be sure to add "crocodilians" to list of well-known vocal species.

2. "innovations in the sound production apparatus among choanates were acquired posteriorly to the first..." – awkward phrasing – just say "acquired after".

Reviewer #3:

Remarks to the Author:

This paper addresses the interesting subject of the evolutionary origin of acoustic communication in vertebrates. Namely, this study focuses on the evolution of acoustic communication in relatively understudied groups (in the context of acoustic communication) such as lepidosauroids, chelonians, non-anuran amphibians, and dipnoi fish. The manuscript is clearly written and well presented. The authors did an impressive job at collecting primary acoustic repertoire and compiling it with data from previously published literature. They conclude that the presence of acoustic communication is a homologous trait in choanate vertebrates and that this supports the hypothesis that the sound apparatus among choanates was acquired prior to the common appearance of acoustic communication.

There are several aspects of the manuscript which may require elaboration or clarification before publication. Below, I provide a list of comments for the authors to consider.

- Introduction

- o 37-46 consider adding a short overview of vertebrate acoustic communication before describing hypotheses for its origin.

- o 44-46 this sentence is unclear. Consider clarifying what phylogenetic reconstruction analyses are and how they fit in the context of acoustic communication evolution before introducing its shortcomings.

- o Following on this point – "species" should probably be "taxa" or "clades". It's not required to have data from particular species to answer the questions in this manuscript, but helps to have data from members of underrepresented clades.

- o 57-59 consider adding context related to vocal production in choanate communication.

- o 63-66 please clarify why certain choanate groups were excluded from the study (e.g., archosaurs, anurans) - while some of this is obvious, it would still help to clarify why they were excluded.

- o 61-68 please clarify the knowledge gap, hypotheses, and predictions that are being addressed or tested.

- o Generally - consider revising the text to avoid over-stating novelty of some of the recordings. For example, while some of the species of chelonians recorded have not been recorded before, others have. The finding that chelonians vocalize is not of itself novel, and at least one of the authors was

instrumental in demonstrating this in a number of papers dating back nearly a decade. So, I wonder if a careful revision of the text can help to more clearly focus on which parts are novel, and which are building on (exciting) previous research.

- Methods

- o 284-291 please clarify what species were recorded. Consider including or citing a supplemental table.

- o 299-301 please clarify which species from Chen and Wiens 2020 were included in the analysis. Consider citing/including a supplemental table.

- o 304-305 same as above, please clarify the species and information collated from other sources.

- o 314 please clarify how a complex vocal repertoire was defined

- o 321-324 – I think I follow the logic, generally, but I am making a lot of assumptions to do so. Can you provide more detail here? These are audible vocalizations, made using lungs and exhalations... why were they not included? How would including them affect your results?

- o 349-354 please clarify what character states were included and how the acoustic properties were included in the analysis.

- Results

- o 100-108 if possible, it would help to clarify specific character traits that explain acoustic communication as a homologous trait (assuming it is more than presence/absence).

- o Figures 1 and 2 are both impressive and very informative. For clarity, it would help to cite the supplemental material that summarises the data in the figure caption (e.g., maximum likelihood values, tables summarizing acoustic properties).

- o Can you rethink your conclusion that acoustic communication is a homologous trait, from the perspective of more clearly supporting it? Presence/absence doesn't really say much about the trait, because acoustic communication is multi-dimensional (ie not a single trait), so the overall result seems like it's not adding a ton to the existing literature. It may be that clarifying the hypotheses and predictions that are being tested will address this confusion.

- o

- Discussion

- o 112-161 The discussion presents a lot of speculation about reasons for acoustic communication as a homologous trait. Consider further incorporating the results of the study into the discussion and clarifying the specific gap that this study fills.

I hope these comments are helpful, and I congratulate the authors on an interesting study.

Reviewer #4:

Remarks to the Author:

Dear Editor,

Jorgewich-Cohen et al. presents a macroevolutionary analysis of the evolution of acoustic communication across tetrapods. This study integrates a substantial new data set of acoustic recordings (mostly of turtles but also of several phylogenetically critical lineages) an updated literature review that focuses on lineages traditionally thought to lack acoustic communication and a data set from a study previously published in Nature Communications (Chen and Weins, 2020). This new analysis substantially changes the conclusions from Chen and Weins by suggesting that acoustic communication evolved in the common ancestor of (essentially) modern tetrapods. I think this is an important new result that is worthy of publication in the journal and am impressed by the scope of the new data set (which also demonstrates the new result that acoustic communication is present across evolutionary tree of turtles--a significant result in its own right though by itself not one that I would regard as sufficiently broad to warrant publication in a journal like Nature Communications). I found the manuscript generally well written and have only a few comments that I would like to see addressed.

1. The title is inaccurate as the analysis does not include the actinoptes which represent half of vertebrates! This is easily fixed and does not take away from the significance of the study.
2. There is really no discussion of why the conclusion of this study differs from the Chen and Weins (2020). On the face of it, the addition of data for 50 species to an original analysis that included ~1800 might not be expected to change the inference of multiple evolutions of acoustic communication in major tetrapod groups to a single origin in the common tetrapod ancestor. However, the authors have really targeted lineages in the tree that are most crucial to our inference of where acoustic communication arose--turtles, tuataras, caecilians. Since the new analysis is tied so closely to the Chen and Weins' data set I would really like to see some discussion of how sensitive our inferences are to the reconstruction of ancestral states in key lineages.
3. The new data set and figures are terrific. It was difficult to determine when the recordings resulted in novel documentation of acoustic communication in a group versus when they corroborated reports from the field or literature. For example, do the authors have the first documented evidence of tuatara acoustic communication? This might be best clarified in the SI but some tabulation of the number of new species that were identified as acoustic communicators from the authors' recordings versus those discovered from the updated lit review would be helpful.

REVIEWER COMMENTS

Reviewer #1 (Remarks to the Author):

This manuscript is an interesting study that incorporates taxa that are generally neglected in the field of acoustic communication to reconstruct the evolution of acoustic behaviors among choanate vertebrates. The methods used are appropriate, the figures are clear and informative, and the study represents an important contribution to the field. I have a few comments that should be addressed in the revisions to this manuscript.

My major concern is that the character states used for the ancestral state reconstruction must be revised to be consistent with the authors' definition of acoustic communication (lines 311-319). Here, the authors indicate that they are considering acoustic signals for which sound energy is generated in the respiratory tract - if this is the case, they should code all species of the salamander family Plethodontidae as not having acoustic communication and revise lines 86-88 of the manuscript.

The point raised here is relevant and brings up a delicate discussion concerning our study. Our results show that acoustic communication is not only a homologous but also conserved behaviour among choanates. The structures of sound production on the other hand, were subject to extensive changes during the evolutionary history of different clades, posteriorly to the appearance of acoustic communication. Birds, for example, developed a syrinx, while *Parasaurolophus* developed a trombone-like crest, and bats and dolphins separately developed structures able to produce ultrasound calls. Following this rationale, Plethodontidae salamanders lost their lungs after the appearance of acoustic communication and the lungless sound production structure may or may not represent a homologous behaviour to sound production from other choanate vertebrates.

Maslin (1950, page 35) stated that: “The squeaking noise produced by plethodont salamanders introduces a third method of sound production— namely, a vibrating valve associated with a resonating chamber. While it is of interest that this sound is produced by salamanders without lungs, the actual production of such sound is dependent simply on an air reservoir, within which the pressure may be raised, and a vibrating valve, through which the air under pressure may be forced”. Although we know for a fact that these salamanders are not using lungs as the motor source of their vocal behaviour, the innovations of their sound production mechanism most likely derived from a lung based acoustic behaviour in their ancestor. Further research would be necessary to answer if the sound production of these salamanders is homologous to the sound production of other choanatians. Although we understand it is more reasonable to consider these behaviours to be the same, in view of that it requires less steps in the evolutionary process, we are unable to confirm it. For this reason, we decided for a more orthodox approach and followed the reviewer’s suggestion to change the character state to absent in *Aneides lugubris* (the only Plethodontidae salamander included in our analysis – Supplementary material 5). We kept Plethodontidae salamanders in the list of vocal species (Supplementary material 1) as for a reference of what is known about sound production among non-anuran amphibians. It is important to highlight that being vocal does not necessarily entail communication and, therefore, we added the Plethodontidae example in the methods section (lines 261 - 262): “We also decided for excluding

lungless salamanders of the Plethodontidae family, as they might have a different, non-homologous method of sound production”.

In any case, most reports of sound production in salamanders remain fairly anecdotal (no data on call types or call parameters) and may more similar to the non-vocal acoustic behaviors described in lines 321-323 as being excluded from this study. It would help if the authors are more clear about their criteria for character state coding to accomodate the diversity of acoustic signal types included in this study.

We agree with the reviewer that many of the reports we found are somewhat anecdotal. Nevertheless, most of them were only included in the list of vocal species (Supplementary material 1), but were not included in the analysis (list of included species is available in Supplementary material 5). We only assigned the state *presence* of acoustic communication to species when we could find a clear report discussing the usage of sounds for communication in the many works we compiled. We included this information in the methods section (lines 272-273): “Only sounds that were considered by literature reports to be used by the studied species in intraspecific communication were considered”.

In the case of salamanders, only three species (excluding *Aneides lugubris*, as discussed in the previous comment) had their character state changed to *present* in our analysis: *Andrias japonicus*, *Salamandra salamandra*, and *Taricha rivularis*, following Maslin (1950) and Duellman and Trueb (1994).

Maslin (1950) compiled information and discussed about the usage of sounds by salamanders. None of these reports, however, present an elaborated description of the repertoires and are mainly opinion based. For this reason, although we think these three species can be included considering our method choices, we ran the analysis without them to test the impact of this change in our results. Results are shown in the figure below.

The exclusion of these species had no relevant impact in the results of ancestral state reconstruction of any node whatsoever. We, however, included a small discussion about the impact of character state changes in key taxa over the overall results.

There are also several species that are notably absent that the authors might consider including in this study, particularly squamates for which evidence of acoustic repertoires have been recorded (e.g., *Liolaemus chiliensis*, Labra et al. 2013; *Anolis grahami*, Milton & Jenssen 1979).

Both *Anolis grahami* and *Liolaemus chiliensis* were included in the list of vocal species (Supplementary material 1), but were not included in the analysis as they are hypothesized to produce their sounds either accidentally or for interspecific communication (hisses and sniffs). We are happy to add any other taxa to the list of vocal species, but we had to constrain our report as basically every species of Lepidosauria produces some sort of hissing sounds. If the reviewer would know of any reports on more complex acoustic behaviours in this clade that were left aside in our study, we will include them.

In addition, we included a new section in the Discussion to clarify the issue of using hissing sounds in this analysis.

Additionally, the manuscript does not make clear the purpose of the second analysis focusing on the turtle genera. How does this analysis address the main hypothesis of the study (the evolution of acoustic communication among choanates), and how does it add to the conclusions drawn using the larger phylogeny with greater taxon sampling?

The second analysis does not change the main conclusion of the work drawn from the analysis with the larger phylogeny, but it strengthens it.

The tree used in this work and previously used by Chen and Wiens (2020) contains only part of the whole diversity of tetrapods. Turtles, for example, are represented by only 16 out of over 360 known species. Smaller clades are represented by very few species because the tree was built taking into account the richness proportionality, as perfectly summarized by Chen and Wiens (2020): “Proportional [richness] sampling is important for analyses relating traits and diversification. The sampling of species within these major clades was also designed to represent major groups (*e.g.*, orders, families), with sampling roughly proportional to their species richness”.

In our second analysis, we built a tree with every turtle genus, promoting a much more refined resolution for this clade. Although it does not change the conclusions, it guarantees that we are not committing similar mistakes from Chen & Wiens. That is, the changes on character state proposed in our study (less than 25 species) were enough to contradict the results found by Chen & Wiens, demonstrating the sensitivity of ancestral state reconstructions to changes in key taxa. By performing a more inclusive analysis, we diminish the chances that our conclusions are being erroneously influenced by key missing taxa, and instead get a more realistic picture of the whole. At least for turtles, tuataras and lungfish, we were able to cover great parts of the diversity. Since only turtles among these three clades are represented by a number of genera, we only performed an extra analysis with this group. Following the reviewer’s concern, we included a paragraph about the impact of key taxa and the relevance of the turtle analysis to the discussion section (lines 120 - 138):

“Nevertheless, the recent growth of evidence for acoustic communication among certain tetrapod groups, commonly considered to be non-vocal, such as aquatic turtles (*e.g.*, 31, 32, and the new data provided by us in this paper), are key in revealing the common ancestry of such behaviour. In fact, including new evidence from only 14 species (12 turtles, tuatara and lungfish) to the analysis proposed by Chen & Wiens (4) was enough to recover opposite results, that were reinforced by the inclusion of data from our critical study of the literature. The sensitivity of ancestral state reconstruction analyses to the character state of key lineages makes a deeper investigation of poorly studied groups imperative.

Knowledge of the natural history of organisms is fundamental in surveys of the macroevolution of certain features. The intensive documentation of vocal communication in turtles in our study is an example. Recordings, observations and subsequent analyses in a phylogenetic framework suggest the homology of vocal communication across turtles and in the last common ancestor of the clade. This result strengthens our broader conclusions on the origins of acoustic communication among choanate vertebrates. Specifically, by evidencing that acoustic communication is widespread and homologous among all turtle genera, we ensure that an analysis with a much less comprehensive sample of this group is trustworthy and not subject to extensive interpretation changes by switching the state character of only few species”.

Furthermore, turtles alone already represent an important group, and the analysis showing that their common ancestor was already producing sounds is of great interest to a young but growing field of research (turtle acoustics).

Given that the vocalization data recorded in the present study are analyzed more in-depth (e.g., with call repertoires fairly well-defined in the supplement), the authors might consider mapping additional character traits onto this phylogeny that more fully characterize the richness of the vocal repertoire.

We thank the reviewer for acknowledging our effort into reporting the acoustic repertoires of the studied species. Nevertheless, we think that the suggestion surpasses the scope of our study as it entails more detailed analysis into the sound types. Furthermore, we do not feel confident doing further comparative analysis with this dataset for two main reasons: 1. Our sampling was opportunistic and includes the species we could get access to while visiting different institutions. This means that for some species we were able to do a more complete sampling of their vocal behaviour by including representatives of different ages and sex, while for others we were only able to record one or two specimens, creating an uneven report of their repertoire. 2. To be able to compare and map acoustic traits in the phylogeny we need a robust homology hypothesis of these traits, which is really hard to establish considering that we have an uneven description of their acoustic behaviour. We hope that this study, once published, will encourage other researchers to look closer into the acoustic repertoire of different species sometimes classified as non-vocal. With future and more detailed data, we will hopefully be able to produce strong homology hypotheses for comparative acoustic work.

Regarding the methods: are recordings made in-air or under-water? The only recording equipment described is Oceanbase, an underwater recorder. There is no information for microphones used to record in air from terrestrial species (notably *Sphenodon punctatus*). Please also provide the dimensions of the plastic pools used for recording, describe any measures used to reduce generation of standing waves and/or noise if used, and indicate the distance between the source and the microphone.

The reviewer is correct. We indeed forgot to include information about the equipment used for in-air recordings, as most of them were made under water. We, therefore, included the following sentence to the methods section (lines 227-229): “In-air recordings were conducted using a Tascam® recorder DR-100MKIII with sensitivity of -115.5 ± 0.5 dB rel 13 mV/uPa \pm 4dB and frequency band of 5Hz - 96kHz.”

We do not provide information about plastic pools’ size and distance between microphone-source, as this varied greatly during our data collection. Many of the species we recorded are rare, endangered and hard to access. For this reason, we had to visit numerous institutions with a variety of conditions to be able to compile an expressive dataset. We were careful to keep the animals in comfortable sized pools (which also varies depending on the species’ size and habits) and tried to reduce noise/interference by turning off sources of noise and electrical current. Ambient noise was also tested by recording in the pools without any animals. We added the following phrase to the methods section (lines 233-234): “We also recorded ambient sound without the presence of any animals in order to account for possible noise/interference.”

The sound analysis described in the methods section (lines 292-296) does not appear to be included in the manuscript results or discussion. How many

individuals per species were recorded, and how many calls per species (and per call type) were analysed?

We included the following phrase to the methods section (lines 238-239): “These parameters were only used as a first description of the repertoires, and were not subjected to any analysis in the present study”. As already discussed in a previous comment, we are unable to perform robust comparative analyses with the present dataset and also believe this surpasses the scope of our study. We provide a table with description of the recorded sounds as a starting point reference database for future works focused on better understanding the repertoires of each of these species (Supplementary material 3). We included, however, information about the number, life stage and sex of specimens recorded in this study.

Several species included here appear not to have a repertoire of call types (e.g., *Rhinoclemmys pulcherrima* and *Chelodina novaeguineae* have only one call type characterized), and therefore (per the authors' definition of acoustic communication in lines 311-323) should be coded as 0 in the character map.

We coded as 1 (presence of acoustic communication) every genus that has been reported to have at least one of its species using sounds for communication. Although the reviewer is right about *Chelodina novaeguineae* and *Rhinoclemmys pulcherrima*, both genera have other species known to produce several sounds: *Chelodina oblonga*, with 17 sounds (Giles et al., 2009) and *Rhinoclemmys punctularia* (present work). Furthermore, we were only able to record one specimen of *C. novaeguineae* and *R. pulcherrima*, while *C. oblonga* and *R. punctularia* had many more specimens during more time of recording, making them much better representatives for their genus.

Although including the species referred to by the reviewer does not change the methods nor conclusions of our study, we decided to include them to the list as our report might encourage other researchers to look into these species as well.

The years for the Wever citations refer to reprints, and should be corrected to reflect the original years of publication (1985 for *The Amphibian Ear*, 1978 for *The Reptile Ear*).

We implemented the suggested corrections.

Reviewer #2 (Remarks to the Author):

Review of Jorgewich-Cohen et al. “Common evolutionary origin of acoustic communication in vertebrates”

This paper represents an excellent idea, a valuable dataset, and a very worthwhile project overall, and I definitely think that with some revisions it should be published in Nature Communications.

However, although there are a number of rather minor changes in wording and focus, there are a few important oversights in the literature review, especially concerning snakes and crocodylians, that will require some additions to the database and thus re-analysis. I also think that the current Discussion is mostly tangential to the main findings of the study, and needs to be re-written.

I am somewhat reluctant to call these “minor revisions”, but will do so in the expectation that the authors will take the advice given below. I am confident that these additions will clearly strengthen the central conclusion: that tetrapod vocal communication represents a homologous trait, present in the common ancestor of all tetrapods (including sarcopterygian fish).

We truly appreciate the inputs given. We provide detailed replies to the comments below.

Major issues:

The discussion and dataset are conspicuously “turtle-centric” and leave out some of the other non-avian reptiles.

1. One group that is conspicuously absent from any mention in the ms. are the crocodylians, which are well-known to have diverse vocalizations and have been well-studied. The database does include Alligator mississippiensis as a vocal species (scored 1 in the supplemental dataset) but many other crocodylian species are also highly vocal and these need to be added for completeness. See:

Dinets, V. (2011). The Role of Habitat in Crocodylian Communication. (PhD). University of Miami, Miami, Florida.

Garrick, L. D., & Lang, J. W. (1977). Social signals and behavior of adult alligators and crocodiles. *American Zoologist*, 17, 225-239.

Reber, S. A., Nishimura, T., Janisch, J., Robertson, M., & Fitch, W. T. (2015). A Chinese alligator in heliox: formant frequencies in a crocodylian. *Journal of Experimental Biology*, 218, 2442-2447.

Vergne, A. L., Avril, A., Martin, S., & Mathevon, N. (2007). Parent–offspring communication in the Nile crocodile *Crocodylus niloticus*: do newborns’ calls show an individual signature? *Naturwissenschaften*, 94(1), 49-54.

Vergne, A. L., Pritz, M. B., & Mathevon, N. (2009). Acoustic communication in crocodylians: from behaviour to brain. *Biological Reviews*, 84(3), 391-411.

We agree that more information can be included in our data compilation (Supplementary material 1), and added the suggest references to the list for completeness.

The reviewer is correct, our dataset is indeed “turtle-centric”. The reason for this focus is that, based on sparse turtle acoustic literature (*e.g.*, Giles et al., 2009; Ferrara et al., 2013) we hypothesized that this behaviour could be widespread among turtles and decided to focus our sampling on this group. We were interested in recording other groups as well, but there were several limitations regarding our study delineation. For example: We wanted to focus on clades that are not commonly in the focus of bioacoustics studies (*e.g.*, caecilians, salamanders, lepidosaurians, turtles and lungfish), which already excludes some groups broadly known to be vocal, such as mammals, birds, frogs and crocodilians. Furthermore, groups such as salamanders and squamates are represented by an overwhelming number of species of which we would be unable to sample in a representative way during the time frame of our research. For this reason, we decided to focus on smaller groups that could be sampled comprehensively and, instead, perform a literature review for those we were unable to record. Therefore, we sampled as many turtles as we could, the tuatara (which represents 100% of living Rhynchocephalia), and one lungfish genus (out of 3 genera). We also managed to get access to one caecilian, a difficult group to investigate.

In any case, squamates were especially complicated in the decision-making of our coding. As nearly all of them produce hissing sounds, we avoided having to make a decision of homologisation based on this. We already had a brief discussion on this in the methods section, but included a broader section in the discussion in order to further clarify this subject.

Moreover, the tetrapod tree used in this work and previously used by Chen and Wiens (2020) contained only one species of crocodilian (*Alligator mississippiensis*) due to the fact that it takes into account the proportionality of each clade’s diversity.

2. As far as I can tell, all the snakes (Serpentes) are currently scored 0 in the current database, but there are various snake species that make vocalizations, including hisses (which are probably universal) and a few that make more interesting tonal calls. See:

Young, B. A. (1991). Morphological basis of "growling" in the king cobra, *Ophiophagus hannah*. *Journal of Experimental Zoology*, 260, 275-287.

Young, B. A., Sheft, S., & Yost, W. (1995). The morphology of sound production in *Pituophis melanoleucus* (Serpentes, Colubridae) with the first description of a vocal cord in snakes. *Journal of Experimental Zoology*, 273, 472-481.

Young’s 1991 paper reports 21 snake species producing sounds. Interestingly, although the pine snake in the 1995 paper does use a larynx, the “vocal cord” equivalent Young reports looks like a novel innovation: relevant to my suggestions concerning the Discussion

Yes, snakes were all coded 0 (absence of acoustic communication) because we were unable to find any evidence that hissing sounds (produced by nearly all snakes) are used in intraspecific communication, so we decided to take this conservative view (strengthening the main inference/hypothesis of our study). Although relevant to the discussion of bioacoustics among vertebrates, hissing sounds were not included in our analysis because we are not able to confirm if this behaviour shares its evolutionary origins with acoustic behaviours associated with intraspecific communication. As a matter of fact, studies conducted by Young (1991, 1995, 1997, 2003) were all focused

on interspecific communication – mainly defensive behaviour. Furthermore, with the exception of *Pituophis melanoleucus* and *Ophiophagus Hannah*, no other snake species has been reported to produce sounds other than hisses, and that overlap with their hearing range. Even in the case of the referred species, reported sounds were only emitted during defensive behaviour. Young (2003, pg. 303) hypothesized that intraspecific acoustic communication may be found in snakes, although it is most probably not a common behaviour in the group: “The relatively low information content in the sounds produced by snakes suggests that these sounds are not suitable for intraspecific communication. Nevertheless, given the diversity of habitats in which snakes are found, and their dual auditory pathways, some form of intraspecific acoustic communication may exist in some species”

We included *Pituophis melanoleucus* and *Ophiophagus hannah* to the list of vocal species in the Supplementary material 1 as a reference for future studies. We, however, did not include these species in the analysis, following our methodology: “Sound communication entails not only that the animal is producing a sound, but also that it has communicative significance. To take a conservative view in ensuring we are comparing homologous types of acoustic communication, we favoured the hypothesis that the presence of a complex repertoire entails communicative meaning and considered only sounds produced by the respiratory tract (excluding scale scratching and tail rattling, for example). [...] Sounds produced during defensive behaviour such as hissing and sniffing in lizards or bellowing in snakes were not considered to be intraspecific acoustic communication and, therefore, were not included. Although these behaviours might have a common origin to the sounds here considered as acoustic communication, we lack evidence to support this claim and opted for a more conservative approach.”

The new section on hissing and sniffing sounds in the discussion hopefully clarify the important issues raised by the reviewer.

References:

Young, B.A., 2003. Snake bioacoustics: toward a richer understanding of the behavioral ecology of snakes. *The Quarterly Review of Biology*, 78(3), pp.303-325.

3. The current Discussion seems quite tangential to the actual data reported here: it focuses almost entirely upon what was NOT studied (namely actinopterygian fish acoustic communication) and very little on the groups actually covered in the database. This needs to be rethought, and rewritten to focus on the data presented in the current study.

In particular, some discussion of the larynx as the (homologous) site of vocal production in all non-avian tetrapods, and at least a sentence or two about the syrinx as an avian “key innovation,” seem mandatory, and should replace some of the current speculation. The larynx is interesting as it is the main site of vocal production in tetrapods (with birds and odontocete cetaceans being major exceptions), but it also has some key innovations in mammals (the presence of a thyroid cartilage):

Fitch, W. T. (2016). Vertebrate bioacoustics: Prospects and open problems. In R. A. Suthers, W. T. Fitch, A. N. Popper, & R. R. Fay (Eds.), *Vertebrate Sound Production and Acoustic Communication* (pp. 297-328). New York: Springer.

Clarke, J. A., Chatterjee, S., Li, Z., Riede, T., Agnolin, F., Goller, F., . . . Novas, F. E. (2016). Fossil evidence of the avian vocal organ from the Mesozoic. *Nature*. doi:doi:10.1038/nature19852

Goller, F., & Larsen, O. N. (1997a). A new mechanism of sound generation in songbirds. *Proceedings of the National Academy of Sciences*, 94(26), 14787.

The current Discussion goes offers speculations about possible common origins between fish and other vertebrates. I think this would be fine as a final paragraph (in a “for future work” vein) but not dominating the whole discussion.

I would however note in this regard that the rhombomeric similarities cited by Bass in the cited papers across quite different vocal production systems, include the avian syrinx, which is well-known to be an novel organ and bird innovation. Thus, this similarity is weak evidence for any homology between tetrapods and fish.

We made substantial changes to the discussion section following the reviewer’s suggestion.

Regarding the comment about the excerpt where we cite the work from Bass, our understanding is that the fact that birds have developed the syrinx innovation does not affect the logic of the argument. The similarities to which Bass refers are in the connections between acoustic behaviour and brain regions that modulate such behaviours, and not to the morphological structures of sound production *per se* (e.g., larynx and syrinx). To clarify this matter, we revised the phrase as to refer to the brain: “which implies a common vocal-sonic central pattern generator in the vertebrate brain (Bass and Baker, 2008; Bass, 2014; Feng et al., 2015; Barkan and Zornik, 2020)”.

4. The claim that turtles or salamanders are “traditionally” thought to lack vocal communication seems overstated.

Regarding turtles first, such references as Auffenberg’s review (not currently cited) make clear that turtles are vocal, though perhaps less so than other reptiles. Auffenberg, W. (1977). Display behavior in tortoises. *American Zoologist*, 17, 241-250.

See also Gans & Maderson’s well-cited review, who clearly note vocalizations in turtles and snakes:

Gans, C., & Maderson, P. F. A. (1973). Sound producing mechanisms in recent reptiles: Review and comment. *American Zoologist*, 13, 1195-1203.

Therefore I’m not sure to what “tradition” the authors refer, but I’d argue that any well-informed bioacoustician is well aware that turtles make sounds (incidentally, so would any reader of D. H. Lawrence’s famous 1921 “Tortoise Poems” which describe how he thought tortoises were mute, until hearing them vocalize!)

The authors should either give some citations to a well-respected source (e.g. textbooks) to support the use of this term, or replace it with “commonly” or “often” thought...

Similar comments could be made regarding salamander voices – there is plenty of previous literature here e.g. reviews in:

Maslin, T. P. (1950). The production of sound in caudate amphibia. *University of Colorado Studies*, 1, 29-45.

Neill, W. T. (1952). Remarks on salamander voices. *Copeia*, 1952, 195-196.

Taxon-specific reports:

Coleman, A. (2016). Sound production in the small-mouthed salamander (*Ambystoma texanum*) (Masters).

Crovo, J. A., Zeyl, J. N., & Johnston, C. E. (2016). Hearing and Sound Production in the Aquatic Salamander, *Amphiuma means* *Herpetologica*, 72(3), 167-173.

Davis, J. R., & Brattstrom, B. H. (1975). Sounds produced by the California newt, *Taricha torosa*. *Herpetologica*, 31, 409-412.

Weber, E., & Schumacher, R. (1975). Der Aufbau der Abwehrrufe des Kammolches (*Triturus cristatus*) und des Fadenmolches (*Triturus helveticus*) (Amphibia, Caudata, Salamandridae). *Salamandra*, 11, 119-129.

Weber, E., & Schumacher, R. (1976). Verschiedenartige Abwehrrufe des Teichmolches (*Triturus v. vulgaris*) (Amphibia, Urodela). *Biologisches Zentralblatt*, 95, 693-701.

Wyman, R. L., & Thrall, J. H. (1972). Sound production by the spotted salamander *Ambystoma maculatum*. *Herpetologica*, 28, 210-212.

We acknowledge that we might have made a poor wording choice in this case and changed the term as suggested. Nevertheless, we would like to provide some explanation. In the case of turtles, some confusion might derive from the fact that species of the Testudinidae are well known to produce various sounds, as stated by Auffenberg (1977) and Gans & Maderson's (1973), since these species live on land and are often kept as pets in many countries. Nevertheless, for aquatic species of turtles comparatively much less data exists regarding their acoustic behaviour. Although we know that sound production is present in quite a few species, recent comparative work has considered turtles to be generally non-vocal (e.g., Chen and Wiens 2020). This idea may have been influenced by some previous literature. A good compilation on this view was written by Russell and Bauer (2021): "Although large numbers of reports of sound emission by turtles are available, only since the turn of the millennium has their sound production been analysed acoustically. Until recently, auditory signals have been deemed to be relatively unimportant for turtles (Harless, 1979; Mrosovsky, 1972) because of their poor hearing abilities (Adrian, Craik, & Sturdy, 1938; Auffenberg, 1977; Pope, 1955). Indeed, Wever (1978) stated that nothing relating to turtle hearing capabilities indicates that sounds serve as useful auditory signals".

Only in the last 20 years, studies have been successively published showing otherwise, starting with Galeotti et al., 2005, Giles et al., 2009, Ferrara et al., 2013 with a few other that followed. In any case, we changed the term *tradition* to "often thought" in order to avoid such confusion.

Regarding salamanders, some species able to produce sounds were already accounted as vocal in the original database published by Chen and Wiens (2020), which we used as the base of our study. Although we made a complementary list of published reports on species not included in their work (supplementary material 1), we did not include the species already assigned as "1" (presence of acoustic communication) by Chen & Wiens to the list (we do provide a list with the character states for each species –

Supplementary material 5). In order to make it clear, we added the following phrase to the legend in the SM1: “Only species that were not accounted as vocal by Chen & Wiens (2020) were included in this list”.

We included the additional species suggested by the reviewer in the revised version. With the new inclusions, one species had its character state changed in the analysis (differed from the state proposed by Chen & Wiens): *Notophtalmus viridescens* (Neill, 1952). For this reason, we ran the analysis again.

The inclusion of this species had no relevant impact in the results of ancestral state reconstruction of any node. We, however, included a small discussion about the impact of character state changes in key taxa over the overall results.

References:

Russell, A. P., & Bauer, A. M. (2021). Vocalization by extant nonavian reptiles: a synthetic overview of phonation and the vocal apparatus. *The Anatomical Record*, 304(7), 1478-1528.

Harless, M. (1979). Social behavior. In M. Harless & H. Morlock (Eds.), *Turtles: perspectives and research* (pp. 475–492). New York: Wiley.

Mrosovsky, N. (1972). Spectrographs of the sounds of leatherback turtles. *Herpetologica*, 28, 256– 258.

Adrian, E. D., Craik, K. J. W., & Sturdy, R. S. (1938). The electrical response of the auditory mechanism in cold-blooded vertebrates. *Proceedings of the Royal Society of London Series B*, 125, 435– 455.

Auffenberg, W. (1977). Display behavior in tortoises. *American Zoologist*, 17, 241-250.

Pope, C. H. (1955). *The reptile world*. New York: Alfred A. Knopf.

Wever, E. G. (1978). *The reptile ear: Its structure and function*. Princeton, New Jersey: Princeton University Press.

Galeotti, P., Sacchi, R., Fasola, M., & Ballasina, D. (2005). Do mounting vocalisations in tortoises have a communication function? A comparative analysis. *The Herpetological Journal*, 15(2), 61-71.

Giles, J. C., Davis, J. A., McCauley, R. D., & Kuchling, G. (2009). Voice of the turtle: the underwater acoustic repertoire of the long-necked freshwater turtle, *Chelodina oblonga*. *The Journal of the Acoustical Society of America*, 126(1), 434-443.

Ferrara, C. R., Vogt, R. C., & Sousa-Lima, R. S. (2013). Turtle vocalizations as the first evidence of posthatching parental care in chelonians. *Journal of Comparative Psychology*, 127(1), 24.

5. There is one published mention of vocal production in African lungfish, along with the South American lungfish, so they could be added to the list: M'Donnell, R. (1860). Observations on the habits and anatomy of the *Lepidosiren annectans*. *Natural History Review*, 7, 93-112.

We were not aware of this incredibly interesting report. It was included to the list and the following sentence was included to the text (lines 86-87): “The African lungfish (*Protopterus annectens*) has been reported to produce sounds (M'Donnell, 1860) and to being able to perceive sounds both in the water and air (Christensen et al., 2015)”.

Minor corrections:

1. Be sure to add “crocodilians” to list of well-known vocal species.

Done. Reports were included to the Supplementary material 1.

2. “innovations in the sound production apparatus among choanates were acquired posteriorly to the first...” – awkward phrasing – just say “acquired after”.

Done.

Reviewer #3 (Remarks to the Author):

This paper addresses the interesting subject of the evolutionary origin of acoustic communication in vertebrates. Namely, this study focuses on the evolution of acoustic communication in relatively understudied groups (in the context of acoustic communication) such as lepidosaurians, chelonians, non-anuran amphibians, and dipnoi fish. The manuscript is clearly written and well presented. The authors did an impressive job at collecting primary acoustic repertoire and compiling it with data from previously published literature. They conclude that the presence of acoustic communication is a homologous trait in choanate vertebrates and that this supports the hypothesis that the sound apparatus among choanates was acquired prior to the common appearance of acoustic communication. There are several aspects of the manuscript which may require elaboration or clarification before publication. Below, I provide a list of comments for the authors to consider.

• Introduction

o 37-46 consider adding a short overview of vertebrate acoustic communication before describing hypotheses for its origin.

Although we agree with the reviewer that a short overview could be nice, the vast literature on this field of research makes it hard to compile relevant and inclusive information in a short statement. Considering that this surpasses the scope of our research, we decided for a more concise version, including a general phrase about the use of acoustic communication by vertebrate animals: “Acoustic communication, broadly distributed along the vertebrate phylogeny, plays a fundamental role in parental care, mate attraction and various other behaviours”. We could, if the reviewer think it is imperative, include a new paragraph focused only on an overview of the area. But we are concerned even this would not do justice to such a broad topic.

o 44-46 this sentence is unclear. Consider clarifying what phylogenetic reconstruction analyses are and how they fit in the context of acoustic communication evolution before introducing its shortcomings.

o Following on this point – “species” should probably be “taxa” or “clades”. It’s not required to have data from particular species to answer the questions in this manuscript, but helps to have data from members of underrepresented clades.

We modified the sentence to better explain what is the objective of the analysis: “Phylogenetic analyses used to reconstruct the ancestral state of acoustic communication along the tree nodes, whilst suggestive of multiple origins, are arguably complicated by missing data from key taxa”.

o 57-59 consider adding context related to vocal production in choanate communication.

We opted for rewriting the discussion in order to give more detail about the contexts of acoustic communication, including different types and mechanisms and, therefore, did not include this in the introduction as to avoid being repetitive.

o 63-66 please clarify why certain choanate groups were excluded from the study (e.g., archosaurs, anurans) - while some of this is obvious, it would still help to clarify why they were excluded.

In our study, we focused our efforts on clades commonly considered non-vocal and present a new dataset on these groups. Nevertheless, we did not exclude any choanate group from the analysis. All groups known to be broadly vocal were included in our phylogeny and dataset – much of the data comes from previous work. To ensure clarity, we included the following sentence (lines 66-70): “Using this new dataset combined with data of well-known acoustic clades (e.g., mammals, birds and frogs), we test if the evolutionary origin of acoustic communication is shared among choanate vertebrates. We suggest a single origin of acoustic communication in the last common ancestor of all Choanata over 400 million years before present (mybp).”

o 61-68 please clarify the knowledge gap, hypotheses, and predictions that are being addressed or tested.

o Generally - consider revising the text to avoid over-stating novelty of some of the recordings. For example, while some of the species of chelonians recorded have not been recorded before, others have. The finding that chelonians vocalize is not of itself novel, and at least one of the authors was instrumental in demonstrating this in a number of papers dating back nearly a decade. So, I wonder if a careful revision of the text can help to more clearly focus on which parts are novel, and which are building on (exciting) previous research.

We framed the introduction so as to explain the contradictions from published works regarding the origins of acoustic communication among vertebrates/tetrapods, therefore addressing the need for a critical view with new data and analyses to address this important question. Further on in the introduction, we mention an alternative hypothesis to the current consensus (acoustic communication is homologous among tetrapods), representing our hypothesis and predictions.

We made alterations in the last paragraph of the introduction in order to clearly state our predictions: “Here, we investigate the evolutionary origins of acoustic communication in choanate vertebrates combining critical new data with phylogenetic trait reconstruction methods using a comprehensive dataset. We assess the acoustic communication abilities in species of diverse vertebrate groups including Lepidosauria (tuataras, lizards and snakes), non-anuran Amphibia (salamanders and caecilians), Chelonians (turtles) and lungfishes (Dipnoi) that are key to mapping vocal communication in the vertebrate tree of life. Using this new dataset combined with data of well-known acoustic clades (e.g., mammals, birds and frogs), we test if the evolutionary origin of acoustic communication is shared among choanate vertebrates. We suggest a single origin of acoustic communication in the last common ancestor of all Choanata over 400 million years before present (mybp)”.

We provide compiled information regarding new and former knowledge on the acoustic behaviour of the groups targeted in this study in supplementary material 1. Regarding turtles, only seven out of the 50 species recorded by us were recorded in previous studies.

- **Methods**

- o **284-291 please clarify what species were recorded. Consider including or citing a supplemental table.**

Done.

- o **299-301 please clarify which species from Chen and Wiens 2020 were included in the analysis. Consider citing/including a supplemental table.**

Done.

- o **304-305 same as above, please clarify the species and information collated from other sources.**

Done.

- o **314 please clarify how a complex vocal repertoire was defined**

Done. We included: “presence of a number of different sounds and/or harmonic calls”

- o **321-324 – I think I follow the logic, generally, but I am making a lot of assumptions to do so. Can you provide more detail here? These are audible vocalizations, made using lungs and exhalations... why were they not included? How would including them affect your results?**

Although relevant to the discussion of bioacoustics among vertebrates, hissing sounds were not included in our analysis because we are not able to confirm if this behaviour shares its evolutionary origins with acoustic behaviours associated to intraspecific communication. They may be a vestigial behaviour of intraspecific communication in groups such as squamates (mostly not able to hear the sounds they produce) that remained and changed its usage to interspecific (e.g., defensive behaviour) during the evolutionary history of this clade. Conversely, they may be a different, non-homologous behaviour, that is instrumentally similar – as it is indeed produced through exhalation, but used in different contexts. Birds, for example, can emit hissing sounds during defensive behaviour but also extensively use complex calls for intraspecific communication (while most squamates do not). Since we have no evidence to answer this homology issue, we favoured a more conservative approach and did not include hissing sounds in our analyses. We included further discussion on hissing sounds in the manuscript to ensure clarity on this issue.

In any case, the inclusion of hissing sounds to the analysis would just strengthen our conclusions. The only main change would be in the squamate node, that would switch from absence to presence of acoustic communication.

- o **349-354 please clarify what character states were included and how the acoustic properties were included in the analysis.**

We apologize, but we are not sure if we follow the question. The character states included in the analyses are stated in this same excerpt: “We reassigned character states based on the information gathered in our literature search and our own recordings: 0 for absence of acoustic communication (which is, in many cases, no more than the absence of information) and 1 for presence. The same analysis was used for the turtle genera

tree. Character states assigned to each species can be found in supplementary tables 5 and 6, respectively”.

- **Results**

- o **100-108 if possible, it would help to clarify specific character traits that explain acoustic communication as a homologous trait (assuming it is more than presence/absence).**

In this study we did not use other characters than presence/absence as it would entail having detailed information on sound types, and this would make such a large-scope study impossible with current knowledge. Our sampling was opportunistic and includes the species we could get access to while visiting different institutions. This means that for some species we were able to do a more complete sampling of their vocal behaviour by including representatives of different ages and sex, while for others we were only able to record one or two specimens, creating an uneven report of their repertoire. Furthermore, to be able to compare and map acoustic traits in the phylogeny we need a robust homology hypothesis of these traits, which is really hard to establish considering that we have an uneven description of their acoustic behaviour. We hope that this study, once published, will encourage other researchers to look closer into the acoustic repertoire of different species sometimes classified as non-vocal. With future and more detailed data, we will hopefully be able to produce strong homology hypotheses for comparative acoustic work.

- o **Figures 1 and 2 are both impressive and very informative. For clarity, it would help to cite the supplemental material that summarises the data in the figure caption (e.g., maximum likelihood values, tables summarizing acoustic properties).**

Done. Thank you for the complement.

- o **Can you rethink your conclusion that acoustic communication is a homologous trait, from the perspective of more clearly supporting it? Presence/absence doesn't really say much about the trait, because acoustic communication is multi-dimensional (ie not a single trait), so the overall result seems like it's not adding a ton to the existing literature. It may be that clarifying the hypotheses and predictions that are being tested will address this confusion.**

As clarified in a reply to a previous comment, we are not able to assume homology of other characters and therefore unable to conduct further analysis on additional traits. Additionally, the conclusions from Chen & Wiens (2020), that used the same approach as here, have been widely cited (53 in the last two years) as the main reference on the subject. Considering that our results are opposite to theirs, and this changes the base ground of many of the following assumptions/interpretations that were published after them, we hope ours is a significant addition to the literature. The new evidence that we provide may serve as the starting point to new research that can look into different aspects of acoustic communication. As we encourage in the new section of our text, hissing sounds can be a good way of looking further into this topic. Furthermore, we now provide evidence that enables further comparative research on choanate vertebrate acoustics, which might have been discouraged in the past due to the former established consensus.

Nevertheless, we changed our discussion section in order to better explain our findings with previously published works that can be interpreted as supporting evidence.

- **Discussion**

- o **112-161 The discussion presents a lot of speculation about reasons for acoustic communication as a homologous trait. Consider further incorporating the results of the study into the discussion and clarifying the specific gap that this study fills.**

We made substantial changes to the discussion section following the reviewer's suggestion, giving more focus to the findings in our study.

I hope these comments are helpful, and I congratulate the authors on an interesting study.

We appreciate the time given by the reviewer to improving our work. We hope you are satisfied with the way we addressed your concerns.

Reviewer #4 (Remarks to the Author):

Jorgewich-Cohen et al. presents a macroevolutionary analysis of the evolution of acoustic communication across tetrapods. This study integrates a substantial new data set of acoustic recordings (mostly of turtles but also of several phylogenetically critical lineages) an updated literature review that focuses on lineages traditionally thought to lack acoustic communication and a data set from a study previously published in Nature Communications (Chen and Weins, 2020). This new analysis substantially changes the conclusions from Chen and Weins by suggesting that acoustic communication evolved in the common ancestor of (essentially) modern tetrapods. I think this is an important new result that is worthy of publication in the journal and am impressed by the scope of the new data set (which also demonstrates the new result that acoustic communication is present across evolutionary tree of turtles--a significant result in its own right though by itself not one that I would regard as sufficiently broad to warrant publication in a journal like Nature Communications). I found the manuscript generally well written and have only a few comments that I would like to see addressed.

1. The title is inaccurate as the analysis does not include the actinopt which represent half of vertebrates! This is easy fixed and does not take away from the significance of the study.

Agreed. We included “choanate” to the title.

We used the term vertebrates as our discussion also included remarks about actinopterygians. Since we did not include them in our analyses and other reviewers suggested we should change some aspects of our discussion and reduce the focus on fish, we agree it is also appropriate to modify the title.

2. There is really no discussion of why the conclusion of this study differs from the Chen and Weins (2020). On the face of it, the addition of data for 50 species to an original analysis that included ~1800 might not be expected to change the inference of multiple evolutions of acoustic communication in major tetrapod groups to a single origin in the common tpod ancestor. However, the authors have really targeted lineages in the tree that are most crucial to our inference of where acoustic communication arose--turtles, tuataras, caecilians. Since the new analysis is tied so closely to the Chen and Wein's data set I would really like to see some discussion of how sensitive our inferences are to the reconstruction of ancestral states in key lineages.

This is a great suggestion. Indeed, some lineages have a greater impact in the reconstruction of ancestral character states than others due to their position in the phylogeny. The widespread presence of missing data for such lineages was the reason for Chen and Wiens to reach conclusions opposite to ours. We included the following to our discussion section (lines 117-128):

“The interpretation of acoustic behaviour as a non-homologous trait proposed in previous research (Russel and Bauer, 2020; Chen and Wiens, 2020) was driven largely by a lack of information on key groups of animals. That is, analyses of ancestral state reconstruction are complicated by the subjectivity of missing data interpretation –

treated as evidence of absence. Nevertheless, the recent growth of overwhelming evidence of acoustic communication among certain tetrapod groups, commonly considered to be non-vocal, such as aquatic turtles (e.g., Ferrara et al., 2014; Lacroix et al., 2022, and the new data provided by us in this paper), are key in revealing the common ancestry of such behaviour. In fact, including new evidence from only 14 species (12 turtles, tuatara and lungfish) to the analysis proposed by Chen & Wiens (2020) was enough to recover opposite results, that were reinforced by the inclusion of data from our critical study of the literature. The sensitivity of ancestral state reconstruction analyses to the character state of key lineages makes a deeper investigation of poorly studied groups imperative”.

3. The new data set and figures are terrific. It was difficult to determine when the recordings resulted in novel documentation of acoustic communication in a group versus when they corroborated reports from the field or literature. For example, do the authors have the first documented evidence of tuatara acoustic communication? This might be best clarified in the SI but some tabulation of the number of new species that were identified as acoustic communicators from the authors recordings versus those discovered from the updated lit review would be helpful.

Most of the species recorded in the present work were done so for the first time, with few exceptions. The tuatara, for example, has been recorded before but with a much simpler description of its repertoire:

Wojtusiak, R.J. and Majlert, Z., 1973. Bioacoustics of the voice of the tuatara, *Sphenodon punctatus punctatus*. *New Zealand Journal of Science*, 16(2), pp.305-313.

Information about the novelty of our dataset can be accessed in Supplemental material 1, where we provide the list of all species recorded in the present work (PW), and the corresponding previous publications on their acoustic repertoire, when existing. Among the 53 species recorded in our study, 41 had no information about their acoustic repertoire whatsoever (considering both sound recordings and anecdotal case reports).

Reviewers' Comments:

Reviewer #1:

Remarks to the Author:

The authors have addressed the concerns I identified in the previous version of this manuscript and I believe the current version is ready for publication.

I commend the authors on their excellent research!

Reviewer #2:

Remarks to the Author:

Review of revision: Jorgewich-Cohen et al.

Overall, this is an excellent revision and the Discussion in particular is greatly improved. I am happy to recommend acceptance, basically as is. Congratulations on a strong paper and great revision!

However there are a bunch of small issues with the Discussion text:

1. The rewording of the Discussion sentence I highlighted still includes a typo: "acquired after to the first"
2. "mostly motored by" should be changed to "powered by"
3. "prevents us to make such statement." Should be "prevents us from making such statement."
4. For the fish homology section, change: "the tetrapod hyoid apparatus" -> the tetrapod hyoid and laryngeal apparatus

Finally, although the decision to exclude snake vocalizations because they are (typically, as far as we know) used inter-specifically in defense makes a certain amount of sense, it is highly conservative. I would suggest that the authors acknowledge this, as a potential topic for future work.

I suggest adding this at line 187 of the revised ms.:

5. "Because our methods exclude many sounds that play a role inter-specifically, e.g. defense hisses by snakes and other species, because they are not known to be used in communication with conspecifics, this is a highly conservative approach. Future studies might broaden this scope by including such calls, since hisses in particular appear to be very common across tetrapods, and are typically defense vocalizations".

Tecumseh Fitch

Reviewer #3:

Remarks to the Author:

Thank you to the authors for their thorough replies to my and the other reviewers' comments. I generally find the manuscript to be improved and I think it will be of interest to readers of Nature Communications.

I see reviewer 1 raised a point about the conditions under which the recordings were made. I had not caught this myself, but I agree with the suggestion to include more information, and I would encourage the authors to revisit this point rather than try to dodge it.

The authors noted that "We do not provide information about plastic pools' size and distance between

microphone-source, as this varied greatly during our data collection. Many of the species we recorded are rare, endangered and hard to access. For this reason, we had to visit numerous institutions with a variety of conditions to be able to compile an expressive dataset. We were careful to keep the animals in comfortable sized pools (which also varies depending on the species' size and habits) and tried to reduce noise/interference by turning off sources of noise and electrical current. Ambient noise was also tested by recording in the pools without any animals. We added the following phrase to the methods section (lines 233-234): "We also recorded ambient sound without the presence of any animals in order to account for possible noise/interference."

I understand that the conditions varied among species and institutions, but this kind of information could still be provided in a supplemental file that would help future work with those species be more directly comparable to your own. I appreciate that producing a table of this information likely feels onerous. However, replication (or even just interpretation of differences among recordings made in different conditions) is impossible without access to these details. As you (the authors) are also aware, factors such as time of day, number of other individuals present, and distance from vocalizing individual to the microphone that really can affect what is recorded. You note repeatedly that this field of study is still growing, and I agree that your analysis is impressive and you make important findings in your paper. You will only strengthen the future value of your work by providing the level of detail that R1 is requesting. Therefore I strongly suggest you consider adding more information that can help your work have greater impact down the road by facilitating comparisons with future recordings from these less-studied species.

Otherwise, I have no major concerns. Thanks again for considering our feedback, and I wish you all the best with your future research.

Reviewer #4:

Remarks to the Author:

The authors have addressed all of my concerns in their revision. I think this is a wonderful study with significant impact for our understanding of the evolution of acoustic communication in vertebrates.

REVIEWER COMMENTS

We appreciate all the suggestions and complements made by the reviewers in this and the previous round of reviews. Your inputs were greatly constructive and contributed to a much better version of this manuscript. Below our responses to each point raised.

Reviewer #2 (Remarks to the Author):

Review of revision: Jorgewich-Cohen et al.

Overall, this is an excellent revision and the Discussion in particular is greatly improved. I am happy to recommend acceptance, basically as is. Congratulations on a strong paper and great revision!

However there are a bunch of small issues with the Discussion text:

1. The rewording of the Discussion sentence I highlighted still includes a typo: “acquired after to the first”

Done.

2. “mostly motored by” should be changed to “powered by”

Done.

3. “prevents us to make such statement.” Should be “prevents us from making such statement.”

Done.

4. For the fish homology section, change: “the tetrapod hyoid apparatus” -> the tetrapod hyoid and laryngeal apparatus

Done.

Finally, although the decision to exclude snake vocalizations because they are (typically, as far as we know) used inter-specifically in defense makes a certain amount of sense, it is highly conservative. I would suggest that the authors acknowledge this, as a potential topic for future work.

I suggest adding this at line 187 of the revised ms.:

5. “Because our methods exclude many sounds that play a role inter-specifically, e.g. defense hisses by snakes and other species, because they are not known to be used in communication with conspecifics, this is a highly conservative approach. Future studies might broaden this scope by including such calls, since hisses in particular appear to be very common across tetrapods, and are typically defense vocalizations”.

Following this recommendation, we included the following statement: “Because we chose for more conservative methods that include only sounds that play a role in communication with conspecifics, and excludes inter-specific communication (*e.g.*, defense hisses by snakes and other species), future studies might broaden this scope by including such calls, since hisses in particular appear to be very common across tetrapods, and are typically defense vocalizations.”

Reviewer #3 (Remarks to the Author):

Thank you to the authors for their thorough replies to my and the other reviewers' comments. I generally find the manuscript to be improved and I think it will be of interest to readers of Nature Communications.

I see reviewer 1 raised a point about the conditions under which the recordings were made. I had not caught this myself, but I agree with the suggestion to include more information, and I would encourage the authors to revisit this point rather than try to dodge it.

The authors noted that "We do not provide information about plastic pools' size and distance between microphone-source, as this varied greatly during our data collection. Many of the species we recorded are rare, endangered and hard to access. For this reason, we had to visit numerous institutions with a variety of conditions to be able to compile an expressive dataset. We were careful to keep the animals in comfortable sized pools (which also varies depending on the species' size and habits) and tried to reduce noise/interference by turning off sources of noise and electrical current. Ambient noise was also tested by recording in the pools without any animals. We added the following phrase to the methods section (lines 233-234): "We also recorded ambient sound without the presence of any animals in order to account for possible noise/interference.""

I understand that the conditions varied among species and institutions, but this kind of information could still be provided in a supplemental file that would help future work with those species be more directly comparable to your own. I appreciate that producing a table of this information likely feels onerous. However, replication (or even just interpretation of differences among recordings made in different conditions) is impossible without access to these details. As you (the authors) are also aware, factors such as time of day, number of other individuals present, and distance from vocalizing individual to the microphone that really can affect what is recorded. You note repeatedly that this field of study is still growing, and I agree that your analysis is impressive and you make important findings in your paper. You will only strengthen the future value of your work by providing the level of detail that R1 is requesting. Therefore I strongly suggest you consider adding more information that can help your work have greater impact down the road by facilitating comparisons with future recordings from these less-studied species.

Otherwise, I have no major concerns. Thanks again for considering our feedback, and I wish you all the best with your future research.

Thank you for this thorough explanation. To address the raised issue, we included information about size and material of enclosures to Supplemental material 3, as requested.

Reviewers' Comments:

Reviewer #3:

Remarks to the Author:

Thank you for adding the information about how the recordings were made. I appreciate you considering the suggestion, and I think this makes the paper much more useful to future researchers. Congratulations on an excellent contribution!

REVIEWERS' COMMENTS

Reviewer #3 (Remarks to the Author):

Thank you for adding the information about how the recordings were made. I appreciate you considering the suggestion, and I think this makes the paper much more useful to future researchers. Congratulations on an excellent contribution!

We appreciate the time and insight given by the reviewer.